# Ultrasensitive Fluorescence Lateral Flow Assay for Simultaneous Detection of *Pseudomonas aeruginosa* and *Salmonella typhimurium* via Wheat Germ Agglutinin-Functionalized Magnetic Quantum Dot Nanoprobe

**DOI:** 10.3390/bios12110942

**Published:** 2022-10-31

**Authors:** Zhijie Tu, Xingsheng Yang, Hao Dong, Qing Yu, Shuai Zheng, Xiaodan Cheng, Chongwen Wang, Zhen Rong, Shengqi Wang

**Affiliations:** 1Beijing Institute of Microbiology and Epidemiology, Beijing 100089, China; 2Medical Technology School, Xuzhou Medical University, Xuzhou 221004, China; 3Science Island Branch of Graduate School, University of Science and Technology of China, Hefei 230036, China; 4College of Life Sciences, Anhui Agricultural University, Hefei 230036, China

**Keywords:** multiplex LFA, magnetic quantum dot, wheat germ agglutinin, *Pseudomonas aeruginosa*, *Salmonella typhimurium*

## Abstract

Point-of-care testing methods for the rapid and sensitive screening of pathogenic bacteria are urgently needed because of the high number of outbreaks of microbial infections and foodborne diseases. In this study, we developed a highly sensitive and multiplex lateral flow assay (LFA) for the simultaneous detection of *Pseudomonas aeruginosa* and *Salmonella typhimurium* in complex samples by using wheat germ agglutinin (WGA)-modified magnetic quantum dots (Mag@QDs) as a universal detection nanoprobe. The Mag@QDs-WGA tag with a 200 nm Fe_3_O_4_ core and multiple QD-formed shell was introduced into the LFA biosensor for the universal capture of the two target bacteria and provided the dual amplification effect of fluorescence enhancement and magnetic enrichment for ultra-sensitivity detection. Meanwhile, two antibacterial antibodies were separately sprayed onto the two test lines of the LFA strip to ensure the specific identification of *P. aeruginosa* and *S. typhimurium* through one test. The proposed LFA exhibited excellent analytical performance, including high capture rate (>80%) to the target pathogens, low detection limit (<30 cells/mL), short testing time (<35 min), and good reproducibility (relative standard deviation < 10.4%). Given these merits, the Mag@QDs-WGA-based LFA has a great potential for the on-site and real-time diagnosis of bacterial samples.

## 1. Introduction

Common pathogenic bacteria pose major threats to human life and food safety because they can cause many serious infectious diseases, including pneumonia, urinary infections, diarrhea, sepsis, and bacillary dysentery [1,2,3]. The World Health Organization estimated that over 550 million people worldwide are infected with bacteria every year and nearly 5.2 million people die from bacteria-related diseases [4,5]. In particular, *Pseudomonas aeruginosa* (*P. aeruginosa*) and *Salmonella typhimurium* (*S. typhimurium*) are two important Gram-negative pathogens widely distributed in natural and hospital environments and responsible for numerous hospitalizations [6,7]. Thus, accurate identification of *P. aeruginosa* and *S. typhimurium* in complex environments is key to guiding timely treatment and saving lives. Early and accurate identification of these bacteria is key to guiding timely treatment and saving lives. Commonly used techniques to detect these pathogens are bacterial isolation and culture methods, polymerase chain reaction (PCR), mass spectrometry, and next-generation sequencing [8,9,10]. However, these methods still have shortcomings, especially in the point-of-care testing (POCT) field, including requirements of a clean room to avoid contamination, tedious operations for sample processing (e.g., cells lysis, DNA extract), long assay time (generally >3 h), sophisticated instruments, and professional manipulators. Thus, a fast, convenient, and sensitive technique for the on-site detection of pathogens must be developed urgently.

Lateral flow assay (LFA), which combines the virtues of chromatographic separation and immune recognition, has become a popular POCT method in biochemical analysis [11,12,13]. The performance of LFA biosensors highly depends on two factors: performance of signal nanomaterials and affinity of used antibodies [14]. In recent years, several types of novel signal nanomaterials, such as surface-enhanced Raman scattering (SERS) nanoparticles, quantum dots (QDs), fluorescent microspheres, and nanozyme nanoprobes, have been integrated into LFA methods as alternatives to traditional colorimetric nanomaterials (mainly colloidal gold) to generate sensitive and readable signals [15,16,17,18,19,20]. Among them, QD materials have demonstrated superior light stability, size-tunable fluorescent emission spectra, high luminescence, and broad excitation spectra and have been proven suitable for the construction of highly sensitive fluorescent LFA biosensors [21,22,23]. However, the application of QD-based LFA methods in multiple bacterial detection has some limitations. Firstly, complicated components in the real bacterial samples (e.g., clinical biological specimens, food matrices) easily affect the stability of QD nanoprobes and, thus, decrease the sensitivity of QD-based LFA. Secondly, one pair of antibodies is usually required for one-target bacterial detection via LFA. To achieve the simultaneous detection of multiple pathogens, the LFA biosensor usually requires multiple types of antibody-modified QD tags to recognize different target bacteria, which greatly increase the difficulty and cost of detection. A QD-based LFA method that can detect *P. aeruginosa* and *S. typhimurium* simultaneously remains to be developed.

Wheat germ agglutinin (WGA) is a well-studied phytolectin that can specifically recognize and bind to certain carbohydrates (mainly N-acetylglucosamine and its derivatives) on the cell wall of most bacteria [24]. Compared with other commonly used biorecognition agents (e.g., antibody, aptamer, bacteriophage, peptide), WGA has obvious advantages of better stability, lower cost, and broad-spectrum binding ability for pathogens [25,26,27]. Moreover, WGA can be modified onto magnetic nanoparticles (MNPs) for the capture of five bacteria in complex samples, with a capture efficiency of more than 70% [28]. Inspired by these findings, we designed a WGA-modified magnetic QD nanoprobe (Mag@QDs-WGA) for the universal capture of *P. aeruginosa* and *S. typhimurium* and introduced it into the LFA biosensor to enable the ultrasensitive and quantitative analyses of the two target bacteria in a single test.

The proposed Mag@QDs-WGA-based LFA has three great innovations compared with previous LFA biosensors for pathogens. Firstly, the Mag@QDs-WGA acts as the universal capture platform for the rapid magnetic enrichment of *P. aeruginosa* and *S. typhimurium* from complex samples, which allows the stable detection of bacteria through the LFA strip. Secondly, the high-performance Mag@QDs nanoprobe is fabricated by coating two layers of carboxylated QDs (CdSe@ZnS-COOH) onto 200 nm Fe_3_O_4_ MNPs, which provides the dual amplification effect of multiple QDs and magnetic enrichment, ensuring the high sensitivity of the proposed assay. Thirdly, the Mag@QDs-WGA-based LFA allows the on-site and simultaneous detection of two target pathogens in 35 min with a low detection limit (25, 28 cells/mL) by constructing two antibody-loaded test lines. To the best of our knowledge, this study is the first to report the direct and multiplex detection of *P. aeruginosa* and *S. typhimurium* via LFA. In addition, the proposed biosensor shows good stability, specificity, and accuracy for the detection of complex bacterial samples (containing food and environment samples). This study suggests that the Mag@QDs-WGA-based LFA has a great potential for POCT applications.

## 2. Materials and Methods

### 2.1. Materials, Chemicals, and Instruments

The materials, chemicals, and apparatures used in this work are described in Appendix A

### 2.2. Synthesis of Mag@QDs Nanobeads

The synthesis of the Mag@QDs nanobeads (Mag@QDs) is shown in Figure 1a. Firstly, Fe_3_O_4_ nanobeads (MB) 200 nm in diameter were fabricated via a classical solvothermal reaction [29]. Then, 50 mL of PEI solution (0.2 mg/mL) was reacted with 1 mL of the prepared MB (10 mg/mL) aqueous solution under 30 min sonication [30]. After magnetic separation, the obtained MB-PEI was redispersed with 50 mL of CdSe/ZnS-MPA QD solution (40 pM) and then sonicated for 40 min. In this step, the CdSe/ZnS-MPA QDs with a negative charge in the solution were tightly packed on the MB surface through electrostatic adsorption, which formed the first QD layer. The fabricated Fe_3_O_4_@QD-shell nanobeads (Fe_3_O_4_@QDs) were magnetically collected, rinsed with deionized water, and then dispersed in 5 mL of ethanol. Afterward, the prepared Fe_3_O_4_@QDs were successively reacted with PEI and CdSe/ZnS-MPA QDs, and Fe_3_O_4_@double QD-shell nanobeads (Mag@QDs) were obtained through PEI-mediated self-assembly. Finally, the obtained Mag@QDs were dried under vacuum at 60 °C, resuspended in 10 mL of absolute ethanol solution, and then stored away from light for further use.

### 2.3. Synthesis of WGA-Modified Mag@QDs

WGA was directly modified onto the Mag@QDs through EDC/NHS chemical coupling. In brief, 1 mL of Mag@QDs (1 mg/mL) was added into a 2 mL EP tube. Then, the Mag@QDs were magnetically separated from the solution and resuspended in 500 µL of MES solution (10 mM, pH 5.5). Subsequently, 5 µL of EDC (0.1 M) and 10 µL of NHS (0.1 M) were added into the mixture and sonicated for 15 min. The activated Mag@QDs were magnetically enriched and resuspended in 200 µL of PBS buffer (0.01 M, pH 7.4). Afterward, 20 µL of aqueous WGA (1 mg/mL) was added to the mixture and incubated for 2 h to form Mag@QDs-WGA. After collecting the production with an external magnetic field, 100 µL of glycine solution (0.75 mg/mL) was added and shaken for another 1 h to block the excess carboxyl group. Finally, the obtained Mag@QDs-WGA was washed twice and stored in 200 µL of PBS buffer (0.01 M, pH 7.4) for further use.

### 2.4. Fabrication of Mag@QDs-WGA-Based LFA for Bacterial Detection

Antibodies to *P. aeruginosa* (1 mg/mL) and *S. typhimurium* (0.8 mg/mL) were coated on the NC membrane by using an XYZ spraying machine (Biodot XYZ5050, Irvine, CA, USA) to form test line 1 (T1) and test line 2 (T2), respectively. The volume of bacterial detection antibody was 20 µL, and the spray rate was set as 1 µL/cm. Next, the antibody-coated NC membrane was dried for 3 h in a 37 °C environment and assembled on the plastic back card of the LFA. Finally, the LFA card consisting of a sample pad, an NC membrane, and an absorption pad were cut into 3 mm-wide LFA strips for subsequent use.

### 2.5. Preparation of Bacterial Sample

The plate culture method was used to quantify the bacteria [31]. The procedures are summarized as follows: *P. aeruginosa* and *S. typhimurium* were inoculated on agar plates containing 5% sheep blood and grown at 37 °C, 5% CO_2_ for 12 h. Then, about 20 colonies were picked from the plate and added into 1 mL of PBS (0.01 M, pH 7.4) as the original bacterial solution. After diluting the bacterial solution 1 × 10^5^–1 × 10^7^ times, 100 µL of the solution was spread on a blood agar plate and incubated as described before. Based on the counting results of colony forming units on the plate, the concentration of the original bacterial solution was calculated and adjusted to the detection concentration.

Then, we investigated the bacterial capture rate of Mag@QDs-WGA in the complex solution. In brief, 4 µL of Mag@QDs-WGA (5 mg/mL) was added into 1 mL of the bacterial sample (10^3^ cells/mL). After 20 min of shaking, the Mag@QDs-WGA-bacterial mixture was magnetically enriched. Then, 100 µL of supernatant liquid was placed on the blood agar plate. After 12 h of incubation, the uncaptured bacteria were counted, and the capture rate was calculated.

### 2.6. Bacterial Detection with Mag@QDs-WGA-Based LFA

Different real samples (i.e., orange juice, cabbage juice, and water) containing 0–10^6^ cells/mL of bacteria were prepared. Orange juice and cabbage juice were purchased from a local supermarket, and river water samples were collected from Summer Palace, Beijing, China. Bacterial detection was conducted as follows. Firstly, 100 µL of concentrated salt buffer (0.1 M PBS, 0.1 M Ca^2+^, 0.1 M Mg^2+^, 0.5% Tween 20) and 4 µL of Mag@QDs-WGA were added into 1 mL of sample solution. After shaking for 15 min, the tube containing mixture was placed on an external magnet. The Mag@QDs-WGA-bacteria complexes in the sample were rapidly enriched to the bottom of EP tube under the action of magnetic field (<2 min), and then the supernatant was removed by a pipette. Subsequently, the complexes were resuspended in 100 µL of running buffer (0.01 M PBS, pH 7.4, 10% FBS, 0.5% milk, 1% Tween 20) and loaded onto the sample pad of the LFA strip. After the 15 min of chromatographic reaction, the test strip was inserted into a portable fluorescence reader (FIC-S1, Suzhou Hemai, China) to simultaneously read the fluorescence signals of two test lines. As shown in Appendix A, the fluorescence signals on the test lines can be obtained within a few seconds.

## 3. Results

### 3.1. Principle of Mag@QDs-WGA-LFA for Bacteria Detection

Figure 1 displays the pathogenic bacterial detection of Mag@QDs-WGA-LFA. Given its broad-spectrum bacterial recognition, strong magnetic enrichment ability, and excellent fluorescence performance, this fluorescence LFA can realize the highly sensitive detection of *S. typhimurium* and *P. aeruginosa* in complex samples. The nanoprobe Mag@QDs-WGA is a key factor affecting the performance of fluorescent LFA. It consists of three components: (i) a 200 nm Fe_3_O_4_ core to ensure strong magnetic responsiveness for rapid enrichment, (ii) two layers of QD-formed shell to provide strong fluorescence intensity and large surface area, and (iii) surface-modified WGA molecules to recognize a broad spectrum of pathogens and effectively bind *S. typhimurium* and *P. aeruginosa* in complex samples.

The inability to detect multiple target antigens simultaneously using a tag modified with one recognition antibody is a common drawback of conventional LFA in bacterial detection because it greatly increases the complexity and production cost of multi-channel detection. In the present study, we introduced WGA to confer the Mag@QDs with a broad spectrum of bacteria-binding ability and simplify the multichannel bacterial detection of LFA. In this study, we aimed to develop a dual-channel LFA biosensor for the simple and highly sensitive detection of *P. aeruginosa* and *S. typhimurium*. The operation principle of the Mag@QDs-WGA-based fluorescence LFA can be divided into two parts. Firstly, the Mag@QDs-WGA universal tags were used to capture a broad spectrum of bacteria in solution and separate them via a magnetic field (Figure 1b). Secondly, the Mag@QDs-WGA-bacteria complexes were resuspended in a running buffer and then transferred to the sample pad of the LFA strip (Figure 1c). The flowing Mag@QDs-WGA-bacteria complexes were captured by the specific antibodies on the test lines and the fluorescence intensities of the test lines were determined by the number of Mag@QDs-WGA-bacteria complexes binding to the test zones. In theory, the higher concentration of target pathogens in the sample, the stronger fluorescence intensities on the corresponding test lines. Finally, quantitative detection of *P. aeruginosa* and *S. typhimurium* was conducted by measuring the fluorescence intensities of the two test lines.

### 3.2. Characterization of Mag@QDs-WGA

Mag@QDs with high fluorescence properties were synthesized according to the PEI-mediated layer-by-layer self-assembly strategy in Figure 1a. Firstly, a Fe_3_O_4_ core was synthesized via a classical solvothermal reaction in which PVP, sodium acetate, and ethylene glycol were used as stabilizer, reducing agent, and solvent, respectively. As shown in Figure 1a, we successfully synthesized 200 nm magnetic cores with uniform particle size and good dispersion. PEI with rich amino groups was used to form a positively charged thin layer on the surface of the Fe_3_O_4_ core, which enabled CdSe/ZnS QDs to be quickly and effectively adsorbed on the Fe_3_O_4_ surface through electrostatic adsorption and amino carboxyl reaction. Figure 1b displays the TEM image of the prepared Fe_3_O_4_@QDs. The negatively charged CdSe/ZnS QDs were quickly adsorbed on the Fe_3_O_4_ core under sonication. The synthesis of Fe_3_O_4_@QDs was characterized by zeta potential. As shown in Appendix A, the zeta potential increased from −39.4 mV to 37.2 mV after the Fe_3_O_4_ core was coated with PEI. After the first layer of QD coating, the zeta potential of Fe_3_O_4_@QDs decreased to 2.4 mV. This result proved the successful adsorption of QDs on the Fe_3_O_4_ surface and provided a basis for the construction of the second set of QD layers. Mag@QDs were successfully constructed using the layer-by-layer adsorption properties of PEI and QDs (Figure 1c). Energy dispersive spectroscopy (EDS) elemental mapping was conducted to characterize the structural composition of the Mag@QDs. As shown in Figure 1d, the elements (Cd, Se, and Zn) of the QD shell were densely distributed on the outer surface of the Fe (green) core, demonstrating the successful construction of the multilayer structure of the Mag@QDs. In addition, the EDS spectrum (Figure 1e) confirmed that the fabricated Mag@QD nanobeads consisted of Fe, Zn, S, Se, and Cd. As revealed in Figure 1f, Mag@QDs in a wide pH range of 3–14 showed stable fluorescence intensity. Moreover, the fluorescence intensity of the Mag@QDs showed no obvious change after 60 days of storage (Figure 1g). Figure 1h shows the fluorescence spectra of the QDs, Fe_3_O_4_@QDs, and Mag@QDs. The fluorescence intensity of Mag@QDs significantly increased to levels 2.41 and 1.52 times higher than those of QDs and Fe_3_O_4_@QDs. The results indicated that the Mag@QD nanobeads have superior fluorescence performance and stability and are suitable for fluorescence-based biosensors.

With EDC/Sulfo-NHS as the activator, WGA can be easily modified to the surface of Mag@QDs. Fourier-transform infrared spectroscopy was used to verify the preparation of Mag@QDs-WGA tags (Appendix A). The magnetic enrichment ability and stability was then verified. As shown in Appendix A, the Mag@QDs-WGA has stable dispersion in complex samples (high salt, orange juice, cabbage juice, and river water) and can be completely separated from all tested samples within 1 min under the action of an external magnetic field. Furthermore, Mag@QDs-WGA maintained a stable fluorescence signal in these complex samples. The results showed that Mag@QDs-WGA can still maintain stable chemical and optical properties in real samples and lay the foundation for application in real samples.

### 3.3. Verification of the Capture Ability of Mag@QDs-WGA for Bacteria

Firstly, the broad-spectrum capture ability of Mag@QDs-WGA for *S. typhimurium* and *P. aeruginosa* was tested. Figure 2a shows the TEM and SEM images of Mag@QDs-WGA binding to bacteria. Mag@QDs-WGA can rapidly and efficiently bind to both *P. aeruginosa* and *S. typhimurium* in samples with 10 s of vortexing. Then, the incubation time and capture efficiency of Mag@QDs-WGA for both bacteria were confirmed through plate counting. As shown in Figure 2b,c, the capture rates of Mag@QDs-WGA for *P. aeruginosa* and *S. typhimurium* reached saturation of 81.64% and 82.62%, respectively, after 15 min of incubation. Appendix A shows the capture efficiency of Mag@QDs-WGA for *P. aeruginosa* at different pH values of 2–12. The results showed that Mag@QDs-WGA exhibited high and stable capture capacity in a wide pH range of 3–11.

WGA is a protein specific for N-acetylglucosamine and its derivatives. Thus, the interference of common monosaccharides on WGA required further investigation. We selected three common sugars (sucrose, glucose, and fructose) as interfering substances. To the best of our knowledge, the content of the three sugars in common food products is lower than 0.2 g/mL. Therefore, we prepared bacterial solutions (10^3^ cells/mL of *P. aeruginosa*) containing 0.2 g/mL of the three sugars as test samples and Mag@QDs-WGA was added to capture bacteria. As shown in Appendix A, the capture efficiency of Mag@QDs-WGA was not affected by these monosaccharides, and WGA showed good carbohydrate specificity. These results suggested that WGA is an efficient and stable broad-spectrum bacterial recognition component that can be applied in fluorescent LFA.

### 3.4. Construction of Mag@QDs-WGA-Based LFA Biosensor

Given that Mag@QDs-WGA is a universal recognition probe, the specificity of fluorescent LFA is mainly dependent on the specific antibodies on the NC membrane. Due to the fact that the proposed Mag@QD-WGA is a typical liquid nanotag, the conjugated pad can be eliminated from the LFA system. The affinity and specificity of the antibody were evaluated by detecting samples containing 10^6^ cells/mL of *P. aeruginosa*, 10^6^ cells/mL of *S. typhimurium*, and 10^6^ cells/mL of their mixture. As shown in Figure 2d, when the sample solution contained both target bacteria, the test line region of the Mag@QDs-WGA-LFA strip showed two bright red lines when irradiated with ultraviolet (UV) light. However, for samples containing *P. aeruginosa* or *S. typhimurium* alone, only one red line was observed on the LFA strip. Figure 2e shows the fluorescent signal values of the test line read by a commercial portable fluorescence strip reader. The results showed no cross-reaction between the two antibodies. Thus, a rapid quantitative assay for the detection of both bacteria could be established.

To realize the best bacterial detection performance of Mag@QDs-WGA-LFA, we optimized the composition of the running buffer, the concentration of the capture antibody, and the immune reaction time of LFA. Different blocking agents in the running buffer directly affect the transport of Mag@QDs-WGA-bacteria complexes and the immune response with capture antibodies. As shown in Figure 2f, Group 4 using a running buffer consisting of PBS (0.01 M, pH 7.4), 10% FBS, 0.5% milk, and 1% Tween 20 can effectively suppress non-specific signals and maximize the signal-to-noise ratio (SNR) value for bacterial detection. The sensitivity of LFA was most affected by the capture antibody on the test line. Thus, the concentration of the capture antibody spray was studied. As displayed in Appendix A, the highest SNR values of the corresponding bacteria were achieved by using 1 mg/mL of anti-*P. aeruginosa* and 0.8 mg/mL of anti-*S. typhimurium*, respectively. Then, we tested the immune response time of LFA because it is a key index for rapid detection. The results in Appendix A demonstrate that 15 min of immune response time was suitable for LFA to produce the highest SNR value on the test line. Notably, setting a control line onto NC membrane for Mag@QD-WGA immobilization is feasible by using anti-WGA antibody. However, considering the high price of anti-WGA antibody, the control line was omitted from the proposed Mag@QD-WGA-LFA strip.

### 3.5. Evaluation of Mag@QDs-WGA-LFA System for Bacteria Detection

The ability of Mag@QDs-WGA-LFA for the ultrasensitive and quantitative detection of both pathogens was validated under optimized conditions. A series of samples containing 10^6^–10 cells/mL of *P. aeruginosa* and *S. typhimurium* was prepared using the gradient dilution method and detected via the established Mag@QDs-WGA-LFA system. Figure 3a,b shows ordinary light and UV light irradiation photographs of the performance of Mag@QDs-WGA-LFA in the simultaneous detection of two bacteria, respectively. The visualization of the test line produced by the Mag@QDs-WGA-bacteria complex gradually weakened as the concentration of the target bacteria decreased. Under normal light conditions, the lowest concentrations of *P. aeruginosa* and *S. typhimurium* observed by naked eyes on the strip were 10^3^ and 5 × 10^2^ cells/mL, respectively. However, the visual sensitivity of the bacteria could reach 50 cells/mL under UV light irradiation. Although the capture rates toward *P. aeruginosa* and *S. typhimurium* were similar, the visual sensitivity to *S. typhimurium* was higher than that of *P. aeruginosa* on the LFA strip, which may be related to the better activity of the *S. typhimurium* antibody than that of the *P. aeruginosa* antibody. Figure 3c shows the heat map of the fluorescence signal with the LFA test line as the source. Figure 3e shows the average fluorescence signal of LFA bands for *P. aeruginosa* and *S. typhimurium* detection after three experiments. When the concentration of the two pathogens decreased to 50 cells/mL, the obtained fluorescence signal values were significantly stronger than those of the blank control group. According to the obtained fluorescence signal and sigmoidal function of bacterial concentrations, the four-parameter logistic fitting curves of *P. aeruginosa* and *S. typhimurium* were drawn (Figure 3f,g). According to the average fluorescence signal intensity of the blank control plus three times the standard deviation of the blank measurement, the limits of detection (LOD) of Mag@QDs-WGA-LFA for *P. aeruginosa* and *S. typhimurium* were calculated to be 25 and 28 cells/mL, respectively. The bacterial concentrations of the two pathogens showed a wide dynamic relationship with the fluorescence signal intensity on the test line. The correlation coefficients of *P. aeruginosa* (R^2^ = 0.996) and *S. typhimurium* (R^2^ = 0.999) ensured the reliability of Mag@QDs-WGA-LFA for quantitative bacterial detection. The precise concentrations of the prepared bacterial samples were determined using plate colony counting. As displayed in Figure 3d, the counting results were consistent with those of Mag@QDs-WGA-LFA, indicating the accuracy of our method. The high capture efficiency of WGA and superior fluorescence performance of the Mag@QDs tag indicated the excellent detection performance of the proposed scheme. Importantly, given the broad-spectrum recognition ability of WGA, only one type of universal nano-probe needs to be added in this protocol (low cost and more convenient). Compared with other LFA-based bacterial detection methods (Table 1) and previously published multiplexed LFA methods (Appendix A) [12,23,32,33,34], the Mag@QDs-WGA-LFA shows significant advantages in sensitivity, universal performance, simplicity of structure, and low cost.

Considering that Mag@QDs-WGA is a universal identification probe, quantification accuracy in dual-channel detection is the most important problem. *P. aeruginosa* and *S. typhimurium* are large (0.5 µm wide and 1–2 µm long), whereas the Mag@QDs-WGA nanoprobe is small (approximately 200 nm diameter). Thus, one separate bacterium can usually bind to multiple nanoprobes. Whether the number of probes we add in can capture all the bacteria in the sample can directly affect the quantitative ability of the dual-channel detection. In the present study, we verified the sensitivity of the single-channel detection of bacteria and compared it with the results of dual-channel detection (Figure 3). As shown in Figure 4a,d, two groups of samples containing different concentrations (0–10^6^ cells/mL) of *P. aeruginosa* and *S. typhimurium* were prepared and detected by Mag@QDs-WGA-LFA. The fluorescent image of the test strips showed that the target bacteria in the solution could produce a red band on the corresponding detection line and had no significant effect on another test line. The visual detection limit of the samples containing only *P. aeruginosa* or *S. typhimurium* was 50 cells/mL. The test lines of the two groups were detected using a portable fluorescence instrument, and the fluorescent signals are shown in Figure 4b,e. The results clearly showed that the fluorescent signal of the corresponding bacteria test line decreased only with the reduction of the target bacterial concentration in the sample. These results confirmed that Mag@QDs-WGA-LFA has good specificity and can be used to detect two target bacteria independently and without interference through two detection lines on an LFA strip. In addition, on the basis of the fluorescent intensity, the corresponding four-parameter logistic equation was drawn, and the LOD was calculated (Figure 4c,f). The LODs of *P. aeruginosa* and *S. typhimurium* were 23 and 25 cells/mL, respectively. The results of the single-channel test (Figure 4) were consistent with those of the dual-channel test (Figure 3), indicating that the number of probes injected was sufficient to capture all the bacteria in the sample. This result also confirmed that the quantitative detection results of the dual-channel LFA were accurate and reliable.

Then, the performance of the LFA based on the WGA proposed in this study was compared with that of the traditional dual-antibodies sandwich-type method. The probe in the LFA of the traditional dual-antibody sandwich method was immuno-Mag@QDs, which were Mag@QDs directly modified with anti-*P. aeruginosa* instead of WGA, and other system parameters were unchanged. As shown in Appendix A, the photo of immuno-Mag@QDs-LFA (0–10^6^ cells/mL) under UV light showed that the minimum visualized concentration of *P. aeruginosa* was 50 cells/mL. Appendix A shows the corresponding fluorescence intensity and calibration curves on the test line, respectively. The LOD of immuno-Mag@QDs-LFA for *P. aeruginosa* was 22 cells/mL, which was close to the detection limit of Mag@QDs-WGA-LFA. The results indicated that WGA can reach the sensitivity of antibodies and can be employed as a low-cost antibody substitute in LFA for bacterial detection.

### 3.6. Specificity and Repeatability of Mag@QDs-WGA-LFA

Reproducibility and specificity are important indexes to evaluate Mag@QDs-WGA-LFA. Figure 5a,c shows the results of five independent tests on 10^6^, 10^4^, and 10^2^ cells/mL of bacterial samples by using Mag@QDs-WGA-LFA. The fluorescence signals of all LFA band test areas were relatively uniform, and their relative standard deviation (RSD) values ranged from 3.96% to 9.46%, indicating the high reproducibility and stability of the proposed method. The spraying antibody ensures the specificity of Mag@QDs-WGA-LFA; thus, it also needed to be verified. We prepared six samples of non-target bacteria, including *Listeria monocytogenes*, *Staphylococcus aureus*, *Escherichia coli*, *Campylobacter jejuni*, *Shigella flexneri*, and *Vibrio parahaemolyticus*, and the same concentration (10^6^ cells/mL) was detected using Mag@QDs-WGA-LFA to evaluate the specificity of the proposed strategy. As shown in Figure 5b,d, no fluorescence signal was found in the negative group containing interfering bacteria, whereas the positive group (including *P. aeruginosa*, *S. typhimurium,* and *P. aeruginosa*/*S. typhimurium* mixture) produced strong fluorescence intensity on the corresponding test line. The results showed that the method has good specificity to target bacteria and strong anti-interference ability to non-target bacteria.

### 3.7. Detection of Real Food and Environment Samples

We then evaluated the detection performance of Mag@QDs-WGA-LFA in real food and environmental samples because *S. typhimurium* and *P. aeruginosa* are mainly transmitted through contaminated food or water sources. Two common food samples (i.e., orange juice and cabbage juice) and one environmental sample (river water) were selected as real samples and added with 1 × 10^6^, 1 × 10^4^, and 1 × 10^2^ cells/mL of pathogens (*P. aeruginosa* and *S. typhimurium*), respectively. Real sample analysis was carried out following the established scheme described in Section 2.6. As shown in Figure 6, Mag@QDs-WGA-LFA obtained relatively stable fluorescence signal intensities in detecting orange juice, cabbage juice, and river water, and these intensities decreased with decreasing bacterial concentrations. The recovery rates of the three real sample models are shown in Appendix A. The average recovery rates of the bacteria in orange juice, cabbage juice, and river water were 94.7–109.5%, 90.2–99.7%, and 93.7–107.3%, respectively, with RSD values below 10.4%. The results showed that the proposed scheme can be applied to complex real samples to detect turquoise and salmon in contaminated food and water resources. Importantly, the proposed Mag@QDs-WGA has a broad-spectrum recognition ability. Thus, this fluorescent probe can be applied to detect other pathogenic bacteria.

## 4. Conclusions

In this study, we reported a two-channel fluorescence LFA biosensor for the simultaneous detection of *P. aeruginosa* and *S. typhimurium* using WGA-modified Mag@QDs as a broad-spectrum capture tool and fluorescent nanoprobe. The WGA-functionalized Mag@QDs nanoprobe showed great advantages over traditional antibody-modified signal tags, including broad-spectrum bacterial recognition, high affinity to bacteria, magnetic enrichment ability, superior fluorescence property, and low cost. By introducing the Mag@QDs-WGA into the LFA system, the proposed biosensor can simultaneously detect *P. aeruginosa* and *S. typhimurium* with LODs of 25 and 28 cells/mL, respectively. Moreover, the established Mag@QDs-WGA-LFA showed high reliability, stability, and accuracy when applied to detect complex bacterial samples. In theory, the detection strategy can be easily applied to other pathogenic bacteria by changing the specific antibodies on the test lines of the LFA strip because of the broad-spectrum recognition ability of WGA. These results indicate that the proposed Mag@QDs-WGA-LFA has great potential to be developed into a powerful tool for the rapid screening of bacteria in field samples.

## Data Availability

Not applicable.

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
