# Peer review of "Ultrasensitive Fluorescence Lateral Flow Assay for Simultaneous Detection of Pseudomonas aeruginosa and Salmonella typhimurium via Wheat Germ Agglutinin-Functionalized Magnetic Quantum Dot Nanoprobe"

_biosensors, 2022, doi:10.3390/bios12110942_

Round 1

Reviewer 1 Report

The paper reports the preparation of LFA systems modified with wheat germ agglutinin modified magnetic quantum dots as a nanoprobe to detect Pseudomonas aeruginosa and Salmonella Typhimurium in food samples. Overall, the paper is well written and well structured. However, there are a few issues and questions which require further clarification by the authors before this paper can be resubmitted:
-    The authors should provide more details about the process for quantum dots magnetical enrichment after contact with bacteria.
-    How did the authors optimize the immobilization conditions of P. aeruginosa and S. typhimurium antibodies onto NC membrane?
-    Authors should provide information about the portable fluorescence spectrophotometer in the experimental section.
-    The authors should discuss the sensing mechanisms involved in the detection of P. aeruginosa and S typhimurium.
-    The authors stated that the proposed sensor exhibited a low cost. However, they did not provide any information regarding the sensor cost. Please provide this information and compare it with other sensing platforms described in the literature as well as with conventional techniques. 

Author Response

Responses to editors and reviewers

Manuscript ID: biosensors-1969023

Title: “Ultrasensitive fluorescence lateral flow assay for simultaneous detection of Pseudomonas aeruginosa and Salmonella typhimurium via wheat germ agglutinin-functionalized magnetic quantum dot nanoprobe”

To referee 1:

Comments:

Reviewer #1: The paper reports the preparation of LFA systems modified with wheat germ agglutinin modified magnetic quantum dots as a nanoprobe to detect Pseudomonas aeruginosa and Salmonella Typhimurium in food samples. Overall, the paper is well written and well structured. However, there are a few issues and questions which require further clarification by the authors before this paper can be resubmitted:

Reply: We are very grateful to the reviewer's professional comments for the manuscript. They are very helpful to our study. According to the constructive advice, we have amended the relevant part in manuscript. Here, the questions are answered below.

Questions:

  1. The authors should provide more details about the process for quantum dots magnetical enrichment after contact with bacteria.

Answer: Thank you very much for your careful review and kind advice. We think this suggestion can make our manuscript much clearer to our readers, so we have added the experimental details about the process for quantum dots magnetic enrichment after contact with bacteria. Please see the red words in in Materials and Methods section 2.6 of the revised manuscript (Page 4).

  1. How did the authors optimize the immobilization conditions of P. aeruginosa and S. typhimurium antibodies onto NC membrane?

Answer: Thank you very much for your careful review and kind advice. In this work, the P. aeruginosa and S. typhimurium antibodies were separately dispensed onto NC membrane by using the XYZ spraying platform (Biodot, USA) at an application volume of 0.1 μL/mm, which is a classical method and widely used in many previous works for LFA strips construction (For example, Wang et al., Anal. Chem. 2020, 92, 15542-15549; Wang et al., ACS Appl. Mater. Interfaces 2021, 13, 40342-40353; Zheng et al., Chemical Engineering Journal 448 (2022) 137760; Xiao et al., Biosensors and Bioelectronics 168 (2020) 112524). The related content about the immobilization conditions of antibodies onto NC membrane can be found in the Materials and Methods section 2.4 of our manuscript (Page 4).

  1. Authors should provide information about the portable fluorescence spectrophotometer in the experimental section.

Answer: Thank you very much for your careful review and kind advice. In this study, we used a portable FIC-S1 fluorescence reader (Suzhou Hemai, China) to simultaneous read the fluorescence signals of test lines. The photograph of portable fluorescence spectrophotometer and its signal output software were shown in Fig. R1. Actually, the information of the used portable fluorescence spectrophotometer has been provided in the Supporting information (S1.2 and Fig. S1) of the original manuscript.

According to the reviewer indicated, we have added the information of portable FIC-S1 fluorescence reader in the experimental section 2.6 to make our manuscript much clearer to our readers. Please see the red words in lines 167-169, page 4.

Fig. R1 (a) Photograph of fluorescence reader. (b) The fluorescence intensity read by the fluorescence reader.

  1. The authors should discuss the sensing mechanisms involved in the detection of P. aeruginosa and S typhimurium.

Answer: Thank you very much for your careful review and professional advice. In this study, we proposed a highly sensitive and multiplex LFA strip for the simultaneous detection of P. aeruginosa and S. typhimurium in complex samples by using WGA-modified Mag@QDs as a universal detection nanoprobe. The Mag@QD-WGA tag with a 200 nm Fe3O4 core and multiple QD-formed shell was introduced into the LFA biosensor for the universal capture of the two target bacteria and provide the dual amplification effect of fluorescence enhancement and magnetic enrichment for ultra-sensitivity detection.

Scheme 1b-c illustrate the sensing mechanisms of the proposed Mag@QD-WGA-LFA for the two target bacteria. The Mag@QD-WGA tags were added into sample solution which contain one or more species of bacteria and incubated for 15 min. During this process, Mag@QD-WGA anchored onto the bacterial surface via strong hydrogen bonding and hydrophobic interactions between WGA molecules and the bacterial cell wall. The formed Mag@QD-WGA-bacteria complexes were then magnetically separated and resuspended in the running buffer. Next, the running buffer containing Mag@QDs-bacteria complexes were directly dropped on the sample pad of LFA to start the chromatographic reaction. During this process, the solution will migrate toward the absorption pad through capillary force, and the formed Mag@QDs-P. aeruginosa or Mag@QDs-S. typhimurium were specifically immobilized by anti-bacterial antibodies on the two test lines (T1 line for P. aeruginosa and T2 line for S. typhimurium) through antibody–antigen reactions. In theory, the higher concentration of target pathogens in the sample, the stronger fluorescence intensities on the corresponding test lines. Finally, ultrasensitive and quantitative detection of P. aeruginosa and S. typhimurium were conducted by measuring the fluorescence intensities of the two test lines using a portable fluorescence reader.

We appreciate the referee’s suggestion very much and have added more content to introduce the sensing mechanisms of our assay for P. aeruginosa and S. typhimurium detection. Please see the red words in page 5.

  1. The authors stated that the proposed sensor exhibited a low cost. However, they did not provide any information regarding the sensor cost. Please provide this information and compare it with other sensing platforms described in the literature as well as with conventional techniques.

Answer: Thank you very much for your careful review and kind advice. There are three reasons demonstrating that the cost of the proposed Mag@QD-WGA-based LFA strip is much lower than that of the conventional Au-based LFA strips.

Firstly, the cost of WGA is much cheaper than that of antibody in every strip. The price of WGA used was 30.68 dollars/mg (https://www.sigmaaldrich.cn/CN/en/substance/lectinfromtriticumvulgariswheat1234598765), and the prices of anti-Pseudomonas aeruginosa and anti-Salmonella typhimurium were 3826.05 dollars/mg (https://www.thermofisher.cn/cn/zh/antibody/product/Pseudomonas-aeruginosa-Antibody-clone-B11-Monoclonal/MA1-83430) and 2993.21 dollars/mg (https://www.abcam.cn/salmonella-typhimurium-lps-antibody-1e6-ab8274.html), respectively. Moreover, 1 mg of WGA can produce 40 mL of WGA-conjugated nanotags (5 mg/mL), which can meet the demand of 10000 test strips preparation. In generally, each AuNPs-based LFA strip uses 0.25 μg antibody for immuno-AuNPs tags fabrication (Parolo et al., Nat Protoc. 2020, 15, 3788-3816; Zhang et al., J. Agric. Food Chem. 2020, 68, 15509). Thus, 1 mg of antibody can only produce immuno-AuNPs for 4000 test strips.

Secondly, each AuNPs-based strip needs 0.0002 mg goat anti-mouse IgG (50 dollars/mg) to build control line, so it costs 0.01 dollar of goat anti-mouse IgG per strip. However, the WGA-based strip does not need goat anti-mouse IgG, thus there is no cost in the goat anti-mouse IgG for the Mag@QD-WGA-based LFA.

Thirdly, the high-performance Mag@QDs nanoprobes were home-made according to the method in our previous studies (Wang et al., Nanoscale, 2020, 12, 795; Wang et al., ACS Appl. Mater. Interfaces 2021, 13, 40342-40353). The cost of 1 g Mag@QDs is about 20 dollars, whereas 1g of AuNPs costs 66.6 dollars.

To sum up, the cost of this method is much lower than that of the conventional Au-based LFA strips.

We appreciate for Editor/Reviewers’ warm work earnestly, and hope that the correction will meet with approval. Once again, thank you very much for your comments and suggestions.

Reviewer 2 Report

It was a pleasure to read this interesting manuscript.

Below typo needs to be corrected before consideration for publication in Biosensors.

1.    Page 9, Line 336: “Considering that Mag@QD-WGA is a universal identification probe, the its quantification accuracy in dual-channel detection is the most important problem”

Author Response

Responses to editors and reviewers

Manuscript ID: biosensors-1969023

Title: “Ultrasensitive fluorescence lateral flow assay for simultaneous detection of Pseudomonas aeruginosa and Salmonella typhimurium via wheat germ agglutinin-functionalized magnetic quantum dot nanoprobe”

To referee 2 :

Comments:

Reviewer #2: It was a pleasure to read this interesting manuscript. Below typo needs to be corrected before consideration for publication in Biosensors.

Reply: We really appreciate your such positive comments! Thank you again for your nice review. According to your comments, we have carefully revised our paper and made some changes. Please see the details below.

Questions:

  1. Page 9, Line 336: “Considering that Mag@QD-WGA is a universal identification probe, the its quantification accuracy in dual-channel detection is the most important problem”

Answer: Thank you very much for your careful review and kind advice. The correction has been made as the referee suggested. In addition, we have proofread the manuscript again and corrected the language mistakes and grammatical errors in the manuscript.

Reviewer 3 Report

The paper “Ultrasensitive fluorescence lateral flow assay for simultaneous detection of Pseudomonas aeruginosa and Salmonella typhimurium via wheat germ agglutinin-functionalized magnetic quantum dot nanoprobe” reports the P.aeruginosa and S. Typhimurium bacteria detection throughout LFA system using a functionalized quantum dot  photoluminescence response. The proposed method for enhanced the adsorption of bacteria on the magnetic quantum dot sound very interesting (using wheat germ agglutinin) since as results report could be useful for other type of bacteria. However some topic need to be improved in the submitted paper:

1.- The authors need to discuss their results comparatively with other  multi LFA system reported comparing the strategy of solutions for example using different layout (matrix like system).

2.- They need to indicate, what is the safe concentration of these bacteria according to the World Health Organization, and discuss if the limit of detection of the proposed device satisfy these condition.

3.- It is necessary to discuss the functionality of the device in relation to the sample preparation procedure in comparison to other LFA devices, since the proposed device eliminates the conjugated pad, and control pad.

Author Response

Responses to editors and reviewers

Manuscript ID: biosensors-1969023

Title: “Ultrasensitive fluorescence lateral flow assay for simultaneous detection of Pseudomonas aeruginosa and Salmonella typhimurium via wheat germ agglutinin-functionalized magnetic quantum dot nanoprobe”

To referee 3:

Comments:

Reviewer #3: The paper “Ultrasensitive fluorescence lateral flow assay for simultaneous detection of Pseudomonas aeruginosa and Salmonella typhimurium via wheat germ agglutinin-functionalized magnetic quantum dot nanoprobe” reports the P.aeruginosa and S. Typhimurium bacteria detection throughout LFA system using a functionalized quantum dot  photoluminescence response. The proposed method for enhanced the adsorption of bacteria on the magnetic quantum dot sound very interesting (using wheat germ agglutinin) since as results report could be useful for other type of bacteria. However some topic need to be improved in the submitted paper.

Reply: We are very grateful to the reviewer's professional comments for the manuscript. They are very helpful to our study. According to the constructive advice, we have amended the relevant part in manuscript. Here, the questions are answered below.

Questions:

  1. The authors need to discuss their results comparatively with other multi LFA system reported comparing the strategy of solutions for example using different layout (matrix like system).

Answer: Thank you very much for your careful review and kind advice. Frankly speaking, this suggestion is really a good one and it helps a lot to make our manuscript be better. We have summarized the specific features of our proposed Mag@QDs-WGA-LFA compared with other previously reported multiplexed LFA methods for bacteria detection in Table S2. More comparison details are also supplemented in the revised manuscript (pages 8-9) by red words.

  1. They need to indicate, what is the safe concentration of these bacteria according to the World Health Organization, and discuss if the limit of detection of the proposed device satisfy these condition.

Answer: Thank you very much for your careful review and kind advice. The two target bacteria Pseudomonas aeruginosa (P. aeruginosa) and Salmonella typhimurium (S. typhimurium) are two important highly pathogenic microorganisms. According to World Health Organization (WHO) regulations, P. aeruginosa and S. typhimurium are not allowed to appear in food and drinking water.

Notably, infections caused by P. aeruginosa/S. typhimurium and other common bacteria (e.g., S. aureus, E. coli O157:H7, Listeria monocytogenes) usually share similar early symptoms (nausea, fever, vomiting, and persistent diarrhea) and are easy to be misdiagnosed. Thus, early and accurate identification of these bacteria is key to guiding timely treatment and saving lives. Traditional diagnostic methods usually need sample enrichment through plate culture to identify the pathogen, but this process takes a long time (12–48 h) even when conducted in specialized laboratories (Charalampous et al., 2019). Modern technologies for bacteria detection include polymerase chain reaction (PCR), DNA sequencing, or mass spectroscopy, which can shorten the testing time and provide accurate results. However, these methods require skilled personnel to operate the sophisticated instruments and clean laboratory space to avoid sample contamination, thus are unsuitable for application in point-of-care testing (POCT).

In this work, we developed a two-channel fluorescence LFA biosensor for the simultaneous detection of P. aeruginosa and S. typhimurium using WGA-modified Mag@QD as a broad-spectrum capture tool and fluorescent nanoprobe. Under the optimal conditions, the established biosensor allowed the multiplex and sensitive identification of the two target pathogens via one LFA strip within 35 min. The LODs of our method reached 25 and 28 cells/mL for P. aeruginosa and S. typhimurium, respectively, which are 11.8 times more sensitive than those from conventional AuNP-based LFA strips and are higher than most of the previously reported LFA methods (Table 1 and Table S2). Thus, we believe that the proposed Mag@QD-WGA-LFA can be valuable in POCT detection of bacteria.

  1. It is necessary to discuss the functionality of the device in relation to the sample preparation procedure in comparison to other LFA devices, since the proposed device eliminates the conjugated pad, and control pad.

Answer: Thank you very much for your careful review and professional advice. In this study, the WGA-modified Mag@QDs was introduced into LFA strip as broad-spectrum capture tool and fluorescent nanoprobe, which can rapidly enrich and separate target bacteria from complex samples without sample pretreatment, not only saving detection time and simplifying the operation process, but also improving the detection sensitivity of LFA. By using Mag@QD-WGA-LFA strip, the rapid detection of bacteria can be achieved in a 1.5 mL EP tube through two consecutive steps (scheme 1b). First, the WGA-modified Mag@QDs were added into sample solution to broad-spectrum capture bacteria, then the formed Mag@QDs-bacteria complexes were completely separated by using an external magnet and resuspended in the running buffer. Second, the running buffer containing Mag@QDs-bacteria complexes were directly dropped on the sample pad of LFA to start the chromatographic reaction. During this process, the solution will migrate toward the absorption pad through capillary force, and the formed Mag@QDs-P. aeruginosa or Mag@QDs-S. typhimurium were specifically immobilized by anti-bacterial antibodies on the two test lines (T1 line for P. aeruginosa and T2 line for S. typhimurium) through antibody–antigen reactions. The fluorescence intensity signals of two test lines were analyzed for the sensitive and quantitative detection of two target bacteria. Due to the proposed Mag@QD-WGA is a typical liquid nanotag, thus the conjugated pad can be eliminated from the LFA system.

In addition, it is indeed that the control line also eliminated from our proposed Mag@QD-WGA-LFA strip. Technically speaking, setting a control line onto NC membrane for Mag@QD-WGA immobilization is feasible by using anti-WGA antibody. As shown in Fig, R2, anti-WGA antibody modified control line can catch the superfluous WGA conjugated Mag@QD, thus can always generate a visible fluorescence band. However, the anti-WGA antibody is difficult to buy and very expensive. For example, the anti-Wheat germ agglutinin antibody (Catalog # ab178444) in this work was purchased from Abcam (Cambridge, UK, https://www.abcam.com), and its price is as high as 682.53 dollars/mg. Actually, if the nanotags of LFA do not use antibody as recognition molecules, the control line of strip is usually not to set in current LFA methods. Many previous works have demonstrated that using other recognition molecules (e.g., antibiotic, phage, peptide, vancomycin) to modify nanotags, the proposed LFA methods possess only test line and no control line is acceptable (For example, “Yue et al., A simplified fluorescent lateral flow assay for melamine based on aggregation induced emission of gold nanoclusters. Food Chem. 2022, 385, 132670”; “Yang et al., Lateral flow assay of methicillin-resistant Staphylococcus aureus using bacteriophage cellular wall-binding domain as recognition agent. Biosens Bioelectron. 2021, 182, 113189”; “Ilhan et al., Replacement of antibodies with bacteriophages in lateral flow assay of Salmonella Enteritidis. Biosens Bioelectron. 2021, 189, 113383; “Zhao et al., Antibiotic and mammal IgG based lateral flow assay for simple and sensitive detection of Staphylococcus aureus. Food Chem. 2021, 339, 127955”).

To sum up, in this work, we want to report using Mag@QD-WGA tag is a good tool for broad-spectrum capture of multiple bacteria and the specific and sensitive detection of different pathogens on the LFA strip. Our results have demonstrated that the proposed LFA with only two test lines can achieve the universal and highly sensitive determination of Pseudomonas aeruginosa and Salmonella typhimurium with low detection limit (25 and 28 cells/mL), short testing time (< 35 min), and high reproducibility (RSD < 9.46 %). Thus, we think omitting the conjugated pad and control line also can manifest the main conclusions of our manuscript and should be accepted.

We appreciate the referee’s suggestion very much and have added more related content to discuss this issue in the revised manuscript (pages 7-8), whicn can make our study more clearer to the readers.

Fig. R2 Photograph and fluorescence intensity of Mag@QD-WGA-LFA strips with control line for P. aeruginosa and S. typhimurium detection.

We appreciate for Editor/Reviewers’ warm work earnestly, and hope that the correction will meet with approval. Once again, thank you very much for your comments and suggestions.
